# A Feasibility Study of Kinematic Characteristics on the Upper Body According to the Shooting of Elite Disabled Archery Athletes

**DOI:** 10.3390/ijerph18062962

**Published:** 2021-03-14

**Authors:** Tae-Whan Kim, Jae-Won Lee, Seoung-Ki Kang, Kyu-Yeon Chae, Sang-Hyup Choi, Yong-Gwan Song

**Affiliations:** 1Korea Institute of Sport Science, Seoul 01794, Korea; burumi@sports.re.kr; 2Department of Special Physical Education, YongIn University, Yongin-si 17092, Korea; spe08@naver.com; 3Graduate School of Education, YongIn University, Yongin-si 17092, Korea; ksk0527@hanmail.net; 4Department of Physical Education and Training, Shanghai University of Sports, Shanghai 200438, China; arin5413@gmail.com; 5Center for Sport Science in Jeju, Jeju 63819, Korea; 6Department of Marine Sports, Pukyong National University, Busan 48513, Korea; 7Department of Marine Design Convergence Engineering, Pukyong National University, Busan 48513, Korea

**Keywords:** disability, ARW2, ARST, tilt angle, trajectory

## Abstract

The purpose of this study is to compare and analyze the kinematic characteristics of the upper limb segments during the archery shooting of Paralympic Wheelchair Class archers (ARW2—second wheelchair class—paraplegia or comparable disability) and Paralympic Standing Class archers (ARST—standing archery class—loss of 25 points in the upper limbs or lower limbs), where archers are classified according to their disability grade among elite disabled archers. The participants of this study were selected as seven elite athletes with disabilities by the ARW2 (*n* = 4) and ARST (*n* = 3). The analysis variables were (1) the time required for each phase, (2) the angle of inclination of the body center, (3) the change of trajectory of body center, and (4) the change of the movement trajectory of the bow center by phase when performing six shots in total. The ARW2 group (drawing phase; M = 2.228 s, *p* < 0.05, holding phase; M = 4.414 s, *p* < 0.05) showed a longer time than the ARST group (drawing phase; M = 0.985 s, holding phase; M = 3.042 s), and the angle of the body did not show a significant difference between the two groups. Additionally, in the direction of the anteroposterior axis in the drawing phase, the change in the movement trajectory of the body center showed a more significant amount of change in the ARW2 group than in the ARST group, and the change in the movement trajectory of the bow center did not show a significant difference between the two groups.

## 1. Introduction

Archery is an event in which victory or defeat is determined by recording scoring points for accurately using a bow and arrow to hit a target at a certain distance [1]. The Paralympic Games started in 1960 in Rome, and the Republic of Korea’s first participation in the Paralympics was at the 1968 Tel Aviv Paralympic Games. Among the events, archery for the handicapped is representative of sportsmen who have achieved excellent results since winning the first gold medal in the Heidelberg Paralympics in 1972 [2]. World Archery defines the functional classification of athletes based on limb impairment. These include two wheelchair classes: Archery Wheelchair 1 (ARW1)—functional impairments in at least three limbs and the trunk and Archery Wheelchair 2 (ARW2)—paraplegia or a comparable disability, as well as an Archery Standing (ARST) class—loss of 25 points in the upper or lower limbs, according to the characteristics of the disability [3,4]. Disability archery, a sport of precision, concentration, and strength open to athletes with a physical disability, is very similar to able-bodied archery, and it is played with records and tournaments. The record matches are played at four distances: 90, 70, 50, and 30 m for men (except the men’s ARW1, which are played at the same distance as the women’s) and 70, 60, 50, and 30 m for women, with 6 points for each of the 6 rounds in 90, 70, and 50 m, as well as the short distance (50 and 30 m), adding to 36 rounds (totally 144 rounds).

Since archery involves using the same posture to aim shots at targets repeatedly, consistency of actions is an essential factor in performance [5]. It also depends on how accurately the drawing and release are carried out, and for experts, there is little variation in the drawing phase. However, for archery players, the most crucial technique is the technology used in the release phase [6]. A previous study on the importance of release in archery reported that the reaction time could be shortened if unnecessary force is not applied during the release, and another study reported that the shorter the clicker’s reaction time in the release operation, the better the performance [7,8]. In addition, for the first time, attempts were made to analyze the consistency of postures in the field of archery, and as a result, posture was reported as being an essential factor to determine the performance when releasing the demonstration from the finger [9]. In archery, motion capture using high-speed cameras is necessary to observe subtle movements and changes in shooting motions. Unfortunately, although there are some studies on general archery players, none of these have analyzed the three-dimensional motion of elite disability archery athletes. Therefore, the present study aimed to: (1) analyze the kinematic mechanisms according to the shooting technique of elite wheelchair archery athletes and provide a reference point for archery athletes with disabilities and (2) find out the movement time, movement of the body, the change of the center of the body, and the bow required to perform the advanced manipulation of archery skills. In particular, the intent is to find out the results of the differences between the two groups, by dividing them into ARW2 and ARST grades, and performing comparative kinematic analysis according to the grades of disabled archery athletes, which is expected to help in knowing the characteristics of each class of athletes more accurately.

Therefore, we hypothesized (1) the time taken for preparing to shoot an arrow would differ between the groups, (2) the angle of body tilt would not show any significant variation between the groups, and (3) the movement trajectories of the body’s center and bow center would vary from phase to phase.

## 2. Materials and Methods

This study included 4 participants from the ARW2 class (sitting in a wheelchair and shooting a bow) and 3 from the ARST class (standing and shooting a bow) and was conducted with the consent of the Korea Archery Association, affiliated with World Archery, which conducts archery for the disabled as well as able-bodied athletes.

For dependent variables related to the shooting, we used a 2 (group: ARW2 or ARST) × 5 (event: E1, E2, E3, E4, or E5) *t*-test analysis. All athletes were selected at the time of the experiment, except ARW1, which has a very high disability level and includes compound items with different bow types. The average player career of the ARW2 group is 15.3 (±5.7) years, and the average player career of the ARST group is 8.6 (±4.0) years. The characteristics of the study participants are shown in Table 1. The arm’s length was measured using a Martin anthropometric device (Tsutsumi, Taitou, Taitouku, Japan), and for precision, an average value was obtained by measuring five times per subject up to zero decimal places.

For data collection, a total of 28 reflective markers were attached to the upper segment and bow using a reflection marker, and the standing calibration was performed to calculate the center of the joint. After shooting a total of 6 shots, the average score was analyzed. The kinematic variables in the experiment were: the time required for each phase, the front and back of the body, the inclination angles of the left and right limbs, and the movement of the upper body and the center of the bow. The knocking part of the bow was set to the center point, the top, and the bottom rim as a vertex, and then the bow center of the three points was defined as the central part of the three points. This study’s experimental equipment consisted of a space coordinate calculation machine, as well as imaging and analysis equipment (Table 2).

The experimental task was to shoot a total of 6 shots at a total distance of 20 m (creating an environment for a 3D motion analysis experiment). The total score of the 6 rounds was 57 and 56 points for the ARW2 and ARST groups, respectively. Its purpose was to calibrate the spatial coordinates necessary for motion analysis in the experimental space until the inclusion of the experimental operation of the subject using the nonlinear transformation technique. As a low-cost alternative to DLT, the proposed nonlinear transformation (NLT) method also uses precision control objects with 3D coordinates of the points where the control objects are required. The reconstruction algorithm uses these points and approximate information about the camera arrangement to solve one camera’s relative orientation using an iterative approach. The scaling and reference frame transformation outside the camera must be defined in the calibration. It used 12 dynamic real-time infrared (I.R.) Eagle 4 cameras (Motion Analysis, Rohnert Park, CA, USA) in the front, rear, left, right, and diagonal directions to analyze the kinematic variables during archery shooting. The center of the body was based on the upper body’s center, excluding the lower body, while the center of the bow was designated as the center of the entire segment’s mass. The sampling rate was 120 Hz, and the camera imaging sensor’s resolution was set to 1280 × 1024 pixels. Additionally, a PD170 3-CCD camcorder (Sony, Minato, Tokyo, Japan) was used to record images using an IEEE-1394 cable, and the shooting speed was set to 30 frames per second (Figure 1). Additionally, the Helen Hayes marker set was used.

The image data were analyzed by using the Cortex 1.3 program (Motion Analysis, Rohnert Park, CA, USA). The events of this study are composed of the events categorized individually based on the classification by Kim et al. [4] (Figure 2 and Figure 3). 

Drawing phase: From the starting point of the drawing to the starting point of the anchoring, which is the moment when the pulling arm started from pulled to stopped completely.Holding phase: From the beginning of holding to the moment when the finger holding the loosen starts releasing.Release phase: From the start of the release to the beginning of the follow-through, which is the moment when the arrow has completely deviated from the bow.Follow-through phase: From the start of the follow-through to its end, i.e., after the end of the rotating bow passes through the waist.

To analyze the change in characteristics of archery shooting behavior of the disabled (ARW2 and ARST), an independent sample *t*-test was conducted using IBM SPSS 25.0 (IBM, Armonk, NY, USA). The significance level was set at *p* < 0.05. To analyze the group’s reliability in 6 shots, the calculation of the coefficient of intraclass correlation, with an average measure of 0.852, was determined to be reliable. Additionally, we reported the 95% confidence interval for the difference and the effect sizes using Hedges’s gav [10].

Approvals were obtained from the ethics committees of the study center (Local Ethics Committee of Institute of Sport Sciences, Ref. 8-B-3727) and other collaborating partners. All the participants provided informed consent.

## 3. Results

The duration of archery shooting was longer in the drawing and holding phases of the ARW2 group than in the ARST group, and the overall shooting time was longer in the ARW2 group. In addition, although there was no difference between the two groups’ body tilt angles, the ARST group showed a smaller change in their body tilt angle. The change in body movement trajectory showed a more significant change in the ARW2 group than the ARST group in the drawing phase’s front and rear directions. Lastly, there was no significant difference in the bow center’s movement trajectory between the two groups.

### 3.1. Time Required for Each Phase

As a result, in the drawing and holding phases, in relation to the time required, the ARW2 group showed a statistically significant difference (*t* = 3.703, *p* < 0.05; 95% CI (0.371, 2.109); Hedge’s *g* = 2.359) compared to the ARST group (*t* = 2.619, *p* < 0.05; 95% CI (0.018, 2.722); Hedge’s *g* = 1.674). In the drawing phase, the ARW2 group required a more extended time than the ARST group, and in the holding phase, too, the ARW2 group took a longer time than the ARST group. Conversely, there were no significant differences in the release and follow-through phases, according to the items (Table 3).

### 3.2. The Body’s Angle of Inclination by Each Event

As a result of the *t*-test on the tilt angle (left and right, anterior and posterior) of the body, there was no significant difference between the two groups (E1 to E5) (Table 4).

### 3.3. Change of the Center of the Upper Body’s Trajectory by Phase

As a result of the *t*-test on the directions of the upper body, X (left and right), Y (forward and backward), and Z (up and down), in each phase, among the kinematic variables of the posture during the archery shooting (*t* = 3.523, *p* < 0.05; 05% CI (−0.115, 1.635); Hedge’s *g* = 2.236), the ARW2 group (M = 1.01 m·s^−1^) showed a more significant *Y*-axis change rate than the ARST group (M = 0.29 m·s^−1^). In contrast, the center-of-gravity movement trajectory’s change in the remainder of the drawing phase, except the *Y*-axis, was not significantly different between the two groups (Table 5).

### 3.4. Change of the Movement Trajectory of the Center of the Bow by Phase

The *t*-test results of X (left and right axis), Y (longitudinal axis), and Z (vertical axis) showed that there were no significant differences between the groups during the archery shots (Table 6).

## 4. Discussion

The purpose of the present study was to examine the kinematic characteristics of elite disabled archers by classifying them into the archery ARW2 and ARST classes. Then, the time required for each phase, the body’s angle of inclination by each event, the change of the center of the upper body’s trajectory, and the change of the movement trajectory of the center of the bow were examined.

The ARW2 group took 1.24 and 1.37 s longer in the drawing and holding phases, respectively, than the ARST group. However, although the difference was not significant, the ARST group took a longer time than the ARW2 group in the release and follow-through phases. Thus, the total duration of the athletes’ performance revealed about 6 s by the shooting behaviors of the elite archery athletes. The holding, release, and follow-through phases averaged 2.53, 0.05, and 0.56 s, respectively [11]. It is reasonable to assume that when drawing, since the support of the forefoot tends to stabilize the body, the ARW2-class athletes (sitting in a wheelchair) will have a longer time for correct preliminary movements. Furthermore, it is thought that they had to make more adjustments when drawing because they lacked lower limb support.

Based on these previous studies, the ARW2 group showed a relatively longer time in the drawing and holding phases, and since its characteristics were significantly lower than the ARST group, it could be expected that more time would have been required to balance the pulling force. In the release and follow-through phases, unlike the previous phase, the ARST group took a longer time than the ARW2 group due to their structural differences (ARST; standing, ARW2; sitting) and how the ARST group sit in a wheelchair. The change in displacement of the hand toward the ground was considerable; yet, there was no significant difference between the two groups in all events. However, previous research indicated that when the body angle was 90 degrees, it was found that the ARW2 group tilted backward more than the ARST group [5].

In archery, the closer the body is to a 90-degree angle, the more stable the posture, by aligning the arm and body skeleton through pushing the bow to reduce the energy and reduce the bow’s shaking by the bow repulsion [12]. Thus, the bodies of the ARST group are considered closer to a 90-degree angle due to low impairment levels (or the advantage of standing motions).

Further, there was no significant difference between the left and right tilt angles. However, the ARW2 group tended to climb higher on the left shoulder than the ARST group. The ARW2 group tilted backward at the trunk’s anterior and posterior tilts, which was connected to the upward movement of the left shoulder. The changes in the center of the upper body’s trajectory showed a significant difference between the two groups in the Y-direction (before and after the trailing axis) in the drawing phase, implying that the body center is more stable in the anterior and posterior direction than that of the ARW2 group. A comparison of center of mass (COM) scores of eight female archery athletes’ good and bad scores showed that the COM score was smaller in a previous study, thereby supporting this and indicating a smaller trajectory of COM movements [5]. As explained earlier, the tilt of the body, as well as the tilt of both shoulders, are also closely related to the upper body’s center.

Conversely, the ARST group showed a smaller COM movement trajectory than the ARW2 group in the remainder of the drawing and holding phases, as well as in the directions *X* (left and right) and *Y*-axis of the release phase. In addition, the *Z* (upper, lower axis) and the follow-through phase of the release phase, as well as the overall direction, showed a more massive shift in the center trajectory of motion, which is due to the upper body’s movement. It is also thought that it was affected by the trajectory. As a precedent study to support this, when an athlete won the gold medal at the Athens Olympic Archery group in 2004, the body center of gravity changes in each phase decreased by 1.21 cm [11].

Finally, there was no significant difference between the two groups in the shift of movement trajectory. It can be seen that the ARW2 group is much lower than the ARST group in the follow-through phase, which is considered to be affected by the bow’s fall, as described above.

At the drawing and holding time, the ARW2 group showed the requirement of a more extended time, but if the time required is shortened, through postural stability training, it will show better performance. Training may be presented for improving postural stability and precision while shortening the time required for coaching staff and athletes. Both groups also showed almost similar kinematic variables, and these data reached the best performing national team players in South Korea. Since it is the players’ data, it will be essential feedback data if the players can be trained.

The more excellent the archery athlete, the higher the body’s stability by using the support of both feet, and the balance training had a positive effect on the record [12,13]. It also uses many of the lower deltoid muscles to minimize the shooting’s shaking and reduces the biceps brachii and triceps brachii [14]. Therefore, for ARW2 players to shoot more stable, training to effectively use the lower deltoid muscles in a situation where they are seated in a wheelchair will be necessary, and repeated continuous practice would be necessary. Previous studies also show that the lower the shoulder joint’s angular velocity is, the higher the recording [6], and training to make the upper limb movement speed in the drawing and follow-through sections similar to those of ARST athletes is necessary.

The limitation of this study is a sufficient number of subjects. Therefore, if research is conducted on a larger number of subjects in the future, higher-quality research results will appear. Additionally, a study that directly compares and analyzes kinematic variables and scores compared to archery performance of general athletes without disabilities should be conducted.

The Kolmogorov–Smirnov and Shapiro–Wilk tests to present detailed data from this study show that the population’s normal distribution is primarily established. Finally, the equivalent variance test of Levene indicated the variable values.

## 5. Conclusions

In conclusion, the characteristics of the kinematic variables between the two groups according to the grade were known. However, since the statistical difference between the two groups was not significant, it should be recognized as a kinematic variable that appears in the event characteristics of the ARW2 and ARST groups. At this point, there is no research on the kinematic characteristics of each sport (grade) of archery for the disabled, so it is judged that a more detailed analysis should be performed for each sport. Further, a study that directly compares and analyzes kinematic variables and scores in relation to the archery performance of general athletes without disabilities should be conducted.

## Figures and Tables

**Figure 1 ijerph-18-02962-f001:**
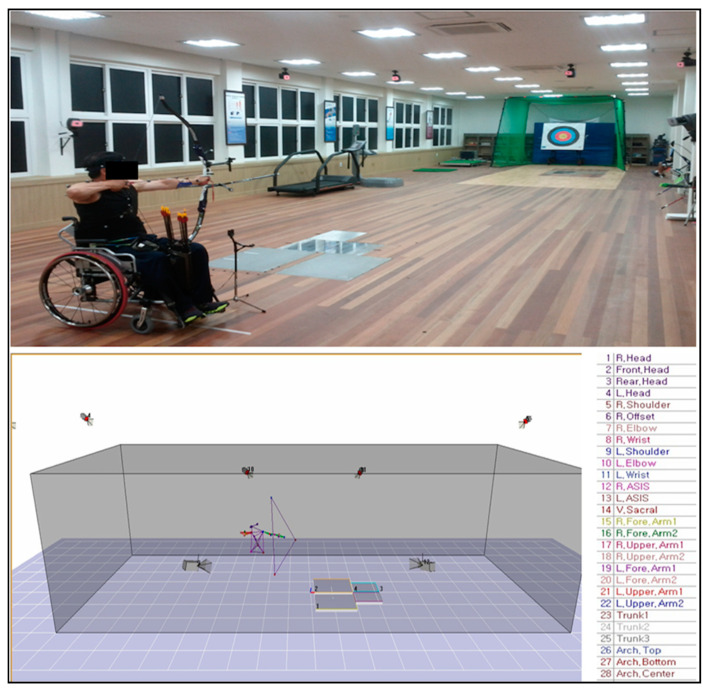
Experimental layout.

**Figure 2 ijerph-18-02962-f002:**
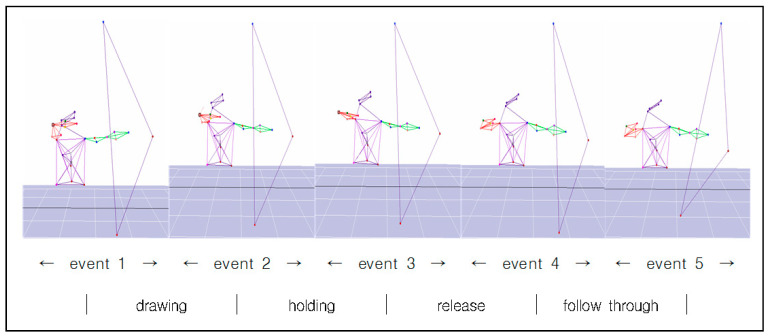
Analytics events and phases. The experimental movement of the study is divided into five events and four phases.

**Figure 3 ijerph-18-02962-f003:**
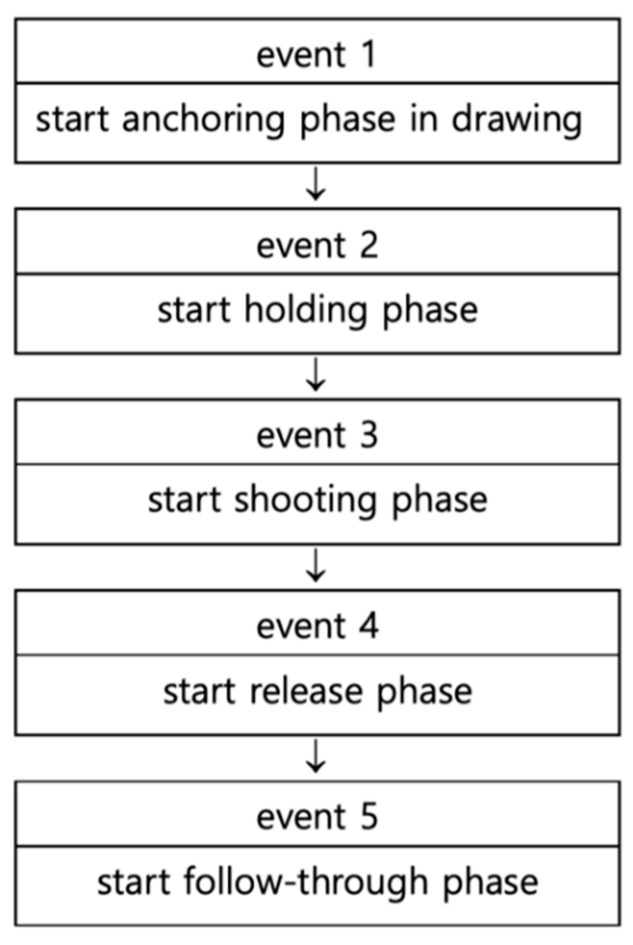
Events flow chart.

**Table 1 ijerph-18-02962-t001:** Characteristics of participants.

Group	Ages (y)	Classification	Length of Arm (cm)
Male	45	ARW2	69.9
Male	43	ARW2	71.1
Male	52	ARW2	74.9
Female	46	ARW2	72.4
Male	51	ARST	69.2
Female	47	ARST	66.0
Female	46	ARST	69.9

ARW2: paraplegia or comparable disability; ARST: loss of 25 points in the upper limbs or lower limbs.

**Table 2 ijerph-18-02962-t002:** Experimental equipment.

Division	Experiment Equipment	Manufacture Company
Calculation and Acquisition of Spatial Coordinates	NLT	Motion Analysis
	Reflection Marker (12.7 mm)	
Motion Shooting	Motion Capture System	Motion Analysis
Motion Analysis	Motion Analysis Software (Cortex 1.3)	Motion Analysis

**Table 3 ijerph-18-02962-t003:** Time required by each phase.

Dependent Variable (s)		Groups	Mean	Standard Deviation	*t*	*p*	Effect Size	95% Confidence Interval of Difference	
								Lower	Upper
Time required by each phase	Drawing	ARW2	2.23	0.52	3.703 *	0.014	2.359	0.371	2.109
		ARST	0.99	0.29					
	Holding	ARW2	4.41	0.85	2.619 *	0.047	1.674	0.018	2.722
		ARST	3.04	0.32					
	Release	ARW2	0.30	0.04	−0.452	0.670	−0.221	−0.130	0.170
		ARST	0.32	0.11					
	Follow Through	ARW2	0.34	0.06	−1.546	0.183	−0.975	−0.118	0.458
		ARST	0.51	0.22					

* *p* < 0.05.

**Table 4 ijerph-18-02962-t004:** The angle of inclination of the body by event.

Dependent Variable (Degree)	Events	Groups	Mean	Standard Deviation	*t*	*p*	Effect Size	95% Confidence Interval of Difference	
								Lower	Upper
The angle of inclination of the body by event (anterior and posterior)	E1	ARW2	70.65	5.12	−1.843	0.124	−1.193	−2.390	14.810
		ARST	76.86	2.94					
	E2	ARW2	70.50	6.22	−1.588	0.173	−1.019	−3.877	16.357
		ARST	76.74	2.89					
	E3	ARW2	70.49	6.94	−1.555	0.181	−0.999	−4.42	17.94
		ARST	77.25	2.97					
	E4	ARW2	69.71	6.97	−1.518	0.19	−0.975	−4.997	19.397
		ARST	76.91	4.86					
	E5	ARW2	69.15	7.35	−1.464	0.203	−0.941	−5.417	19.757
		ARST	76.32	4.66					
The angle of inclination of the body by event (left and right)	E1	ARW2	8.73	2.33	0.572	0.592	0.367	−5.179	8.139
		ARST	7.25	4.54					
	E2	ARW2	9.70	5.89	0.26	0.805	0.169	−9.160	11.240
		ARST	8.66	3.93					
	E3	ARW2	9.80	4.66	0.244	0.817	0.157	−8.114	9.814
		ARST	8.95	4.42					
	E4	ARW2	7.97	3.63	0.046	0.965	0.029	−7.238	7.498
		ARST	7.84	3.93					
	E5	ARW2	9.55	2.28	−0.045	0.966	−0.028	−6.841	7.081
		ARST	9.67	4.86					

**Table 5 ijerph-18-02962-t005:** Change of trajectory of the body center.

Dependent Variable (m·s^−1^)		Groups	Mean	Standard Deviation	*t*	*p*	Effect Size	95% Confidence Interval of Difference	
								Lower	Upper
Change of trajectory of body center	drawing_X	ARW2	1.86	0.37	2.297	0.07	1.435	−0.115	1.635
		ARST	1.10	0.54					
	drawing_Y	ARW2	1.01	0.29	3.523 *	0.017	2.236	0.188	1.252
		ARST	0.29	0.24					
	drawing_Z	ARW2	5.52	1.78	0.972	0.376	0.624	−2.161	4.781
		ARST	4.21	1.75					
	holding_X	ARW2	0.74	0.10	0.38	0.72	0.263	−0.159	0.219
		ARST	0.71	0.09					
	holding_Y	ARW2	0.38	0.31	0.784	0.469	0.522	−0.369	0.709
		ARST	0.21	0.21					
	holding_Z	ARW2	1.32	1.62	0.221	0.834	0.146	−2.370	2.830
		ARST	1.09	0.67					
	release_X	ARW2	1.46	0.35	0.098	0.926	0.083	−0.563	0.623
		ARST	1.43	0.21					
	release_Y	ARW2	0.30	0.19	0.5	0.638	0.294	−0.277	0.397
		ARST	0.24	0.14					
	release_Z	ARW2	0.99	0.15	−0.884	0.417	−0.570	−0.683	1.403
		ARST	1.35	0.82					
	Follow through_X	ARW2	1.73	0.12	−1.282	0.328	−0.985	−1.091	4.311
		ARST	3.34	2.17					
	Follow through_Y	ARW2	0.65	0.43	−0.937	0.392	−0.604	−0.485	1.045
		ARST	0.93	0.32					
	Follow through_Z	ARW2	2.27	0.65	−1.368	0.23	−0.877	−0.796	2.596
		ARST	3.17	1.11					

* *p* < 0.05.

**Table 6 ijerph-18-02962-t006:** Change of the movement trajectory of the bow center by phase.

Dependent Variable (m·s^−1^)		Groups	Mean	Standard Deviation	*t*	*p*	Effect Size	95% Confidence Interval of Difference	
								Lower	Upper
Change of the movement trajectory of the bow center	drawing_X	ARW2	0.63	0.24	1.489	0.197	0.941	−0.182	0.662
		ARST	0.39	0.17					
	drawing_Y	ARW2	0.25	0.15	0.312	0.768	0.199	−0.219	0.279
		ARST	0.22	0.08					
	drawing_Z	ARW2	0.92	0.33	1.436	0.21	0.942	−0.257	0.937
		ARST	0.58	0.26					
	holding_X	ARW2	0.21	0.08	−1.305	0.249	−0.754	−1.279	1.340
		ARST	0.27	0.04					
	holding_Y	ARW2	0.11	0.10	0.151	0.886	0.100	−0.154	0.174
		ARST	0.10	0.05					
	holding_Z	ARW2	0.17	0.14	0.019	0.985	0.000	−0.241	0.241
		ARST	0.17	0.09					
	release_X	ARW2	0.31	0.30	0.706	0.511	0.467	−0.330	0.590
		ARST	0.18	0.05					
	release_Y	ARW2	0.20	0.15	0.91	0.405	0.637	−0.144	0.324
		ARST	0.11	0.04					
	release_Z	ARW2	0.24	0.10	0.081	0.939	0.063	−0.250	0.270
		ARST	0.23	0.17					
	Follow through_X	ARW2	0.42	0.25	−0.813	0.453	−0.494	−0.305	0.565
		ARST	0.55	0.17					
	Follow through_Y	ARW2	0.23	0.20	−0.292	0.782	−0.182	−0.403	0.503
		ARST	0.28	0.27					
	Follow through_Z	ARW2	0.09	0.05	−0.969	0.432	−0.726	−0.268	0.688
		ARST	0.30	0.38					

## Data Availability

The data that support the findings of this study are available from the corresponding author upon reasonable request due to ethical and privacy restrictions.

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
