# Peer review of "A Feasibility Study of Kinematic Characteristics on the Upper Body According to the Shooting of Elite Disabled Archery Athletes"

_ijerph, 2021, doi:10.3390/ijerph18062962_

Round 1

Reviewer 1 Report

I would like to commend the authors on their work and I believe the revisions made have greatly improved the quality of the paper. This was a well-done study with a unique population bringing a great deal of originality and novelty. I have no further critiques to the paper, as it currently stands. Best of luck with the rest of the review process.

Author Response

Infinite thanks for complimenting our thesis. We have tweaked many parts in other reviews, and we have no doubts that we will produce quality results. Once again, thank you for your review of this paper.

Reviewer 2 Report

This manuscript describes the upper-body kinematics during archery shooting in two Paralympic archery classes. Biomechanical studies of elite-level athletes are generally rare, especially in Paralympic populations, and should therefore be welcomed. The authors should be commended for an unconventional investigation. However, there are several comments and suggestions, as outlined in the attached review, that I would strongly encourage the authors to consider.

Reviewer 3 Report

The aim of this study is to compare and analyze the kinematic characteristics of the 14 upper limb segments during the archery shooting in two Class. I commend the authors on this great research manuscript. The topic of investigation is interesting for both practice and sport science. However, I feel this paper need further improvement before publication.

Title

Due to the small size of the sample, I consider it more interesting to include the manuscript as a "pilot study"

Abstract

Line 22. Please, include values and its significant (p value).

Method

Line 95. Please, manufactured, city and country of the camaras.

Line 100. Please, manufactured, city and country of the camcorder.

Consider including also a sequencing of motion pictures below figure 2. It is not possible to correctly observe the movement in the current figure 2.

In the materials and methods section, it is recommended that you begin with a precise definition of the research design you employ.

The authors can include statistical tests to know the normality and homogeneity of the data.

Please, include the effect size and coefficient interval. Also, in results.

Results

Table 3,4 and 5. Including a new column with the values of the size of the effect and its confidence interval. Also, include a legend with the meanings of the acronyms (E1, E2, E3…).

Table 5 and results, please change “m/s” by ”m·s-1

Discussion

Please, at the end of the discussion include the limitations of the study.

References

Please, adapt references to the style of the journal

Round 2

Reviewer 3 Report

Thanks for taking some of my comments and revising the paper. However, some of my comments cannot be addressed clearly. Further revision and clarification might be required before publication.

Why the authors decided to use the T-test for independent samples instead of Mann-Whitney U? I once again indicate the need to include the tests of normality and homogeneity used.

Due to the sample size, use Hedge's g instead Cohen’s d.

Author Response

This study should use the Mann-Whitney test because you say that less than ten samples. However, we wanted to mark the average and standard deviation to present detailed data in this study and that the normal distribution of the population was mostly established through Kolmogorov-Smirnov and Shapiro-Wilk tests. Therefore, it would be appreciated by a very lacking study, but we would like to understand the current data presentation law to improve subsequent studies. The relevant contents were described in the text of the study (L 275-277).

Also, Cohen's d value was changed to the Hedge's g value.

Although we lack much research, our research has improved, and we learned a lot thanks to your faithful review. It is a great honor.

This manuscript is a resubmission of an earlier submission. The following is a list of the peer review reports and author responses from that submission.

Round 1

Reviewer 1 Report

Brief summary

The authors conducted a pilot observational study in n = 7 elite disabled archers with the primary aim of studying differences in timing/kinematic characteristics during shooting between archers in a standing position (n = 3) and in a sitting position on a wheelchair (n = 4). Predominantly, no significant differences between groups were found; however, significantly longer drawing and holding time were reported in sitting archers compared to standing archers. The authors concluded “that kinematic performance according to each group was, in that there was a measurable variation in kinematic variables” (?) and “the overall shooting time was longer in the ARW2 group”.

Broad and specific comments

Thank you for the opportunity to review the manuscript and for the work conducted by the authors. Strengths of the study include elaborate data collection and a rarely studied sample. However, having said this, I have severe concerns regarding the manuscript. Therefore, I cannot recommend publication of the manuscript. Some of my concerns, but not all, are outlined below.

  1. The introduction of the manuscript does not lead to the aim of the study; presumably due to the following issues: a) the aims of the study (stated in the second last paragraph of the introduction) are not in line with the hypotheses (stated in the last paragraph of the introduction). While the aims refer to a descriptive question unrelated or at least less related to group comparisons, the hypotheses include a group comparison. b) The rationale of the manuscript is missing. I cannot see the importance of comparing the duration of different phases during shooting of standing and sitting archers as outlined in the manuscript. Thereby, the guideline for the introduction in the Instruction for authors may be considered as not fulfilled (“The introduction should briefly place the study in a broad context and highlight why it is important. It should define the purpose of the work and its significance…”).
  2. Methods
    1. Although I understand that the recruitment of disabled athletes is difficult, the small sample size (that is never stated as a limitation in the manuscript) severely limits the informative value of the study.
    2. Inferential statistics are potentially biased since a parametric test is used. As a minimum, tests on normality should be conducted and, if normality cannot be assumed, a non-parametric test should be used and reported.
    3. Resolution of Figure 1 does not allow to read/allocate the information provided.
  3. Results
    1. Partly connected to the small sample size, it is unclear, how the significant group differences have to be interpreted. For instance, the differences might simply reflect sex differences, as 25% of the sitting archers are female in contrast to 67% of the standing archers. But also, other explanations are valid, e.g., it is unclear if a longer drawing/holding time is (dis)advantageous during sitting compared to standing. However, a connection of the time/kinematic variables to the precision is missing. It seems important to take into account the precision of the shot to draw meaningful conclusions.
    2. Information is provided both in Tables and Figures. Usually, the most appropriate form of presentation should be used, and redundancies should be avoided.
    3. Units of the variables should be provided for all Tables/Figures.
    4. Error bars in the Figures should be explained.
  4. Although the scope of the present Journal is relatively wide, I doubt that the content of the manuscript fits the scope of the journal/section, thereby limiting the attraction to the readers. A journal with a stronger focus on biomechanical aspects and/or elite athletes might be a more appropriate option.
  5. Minor issues:
    1. Language corrections/checks on redundancy seem necessary, e.g., Page 2, 1st paragraph: “Also attempted to analyze the…”, page 8, last paragraph “The ARW2 group had 1.243 seconds longer…”, page 9: “The results of this study are as follows. First, the results of this study are as follows”
    2. Wording of some expressions have to be questioned, e.g., page 3, “The purpose of this study was…”, page 9, last paragraph of the discussion: “that the ARW2 group is significantly smaller…”, Abbreviation COM is not explained
    3. Consistent formatting should be provided, e.g., page 6, title of Figure 4.

Reviewer 2 Report

First and foremost, I would like to commend the authors on a well-written paper. The primary critique I have is the lack of a practical applications section. How can the information gained from this study be used to enhance sport performance. If you were to explain this study back to the athletes and the coaching staff, how would they utilize the results. Additionally, what are some future directions you could take this research based on what was observed. The discussion and conclusion sections seem extremely "light". 

Reviewer 3 Report

The authors consider the change in the kinematics of the upper body during the archery shooting of Paralympic Wheelchair Class archers and Paralympic Standing Class archers. The authors found no significant difference in the mean between the two groups. Control group missing.

Unfortunately, a big drawback of the manuscript is the lack of data on changes in the activity of skeletal muscles, which does not allow discussing possible mechanisms for controlling voluntary movements (posture) of a person, which should determine the result.

Round 2

Reviewer 1 Report

This is the revised version of a manuscript I was invited to review earlier. Thank you for the opportunity to again review the manuscript. The (very few) changes conducted by the authors did not change my severe concerns regarding the manuscript. Therefore, I cannot recommend publication of the manuscript. Some of my concerns, but not all, are outlined below.

Referring to my previous comments and additional comments

  1. Neither the aims of the study nor the hypotheses changed in the present version and my impression is that it needs more than 2 non-literature-based added sentences. Therefore, the aims and hypotheses are still not in line.
  2. Methods
    1. No answer was given to my previous comment, i.e., the small sample size. Connected to the small sample size, inappropriate conclusions on non-significant finding are done (“However, the statistical difference between the two groups was not significant, so it should be recognized as a kinematic variable that appears in the ARW2 and ARST groups' event characteristics.”). Absence of evidence is not evidence of absence…
    2. No changes on the statistical analysis section done.
    3. Resolution of Figure 1 does not allow to read/allocate the information provided.
  3. Results
    1. No answer was given to my previous comment (“Partly connected to the small sample size, it is unclear, how the significant group differences have to be interpreted. For instance, the differences might simply reflect sex differences, as 25% of the sitting archers are female in contrast to 67% of the standing archers. But also, other explanations are valid, e.g., it is unclear if a longer drawing/holding time is (dis)advantageous during sitting compared to standing. However, a connection of the time/kinematic variables to the precision is missing. It seems important to take into account the precision of the shot to draw meaningful conclusions.”)
    2. The results section (still) contains imprecisions, e.g., l. 137 (“<.05 and t = 2.971, p <.05 to the items. ARW2 group (M = 2.3) (M = 0.98).” -> ARST missing?) and inconsistencies (l. 136 and l. 149, t-, T- statistics).
    3. Units are still missing in the main text (e.g. l. 137 ff.).
  4. Some minor issues were not corrected (e.g., l. 54).

Reviewer 3 Report

The quality of the Figure has been greatly improved.

However, there is a lot of speculation about the results (.Also, when drawing, the support of the forefoot plays a role in stabilizing the body). Where is the confirmation of this.

It is necessary to show the electromyogram pattern changes for confirmation.

Next. "After all, they had to adjust more when drawing because they lack the support of the lower extremities" . This speculation is not confirmed by anything
